# Inhibition of Chikungunya virus nsP2 protease *in vitro* by scorpion venom peptide pantinin-1

**Mohammadamin Mastalipour[1], Mônika Aparecida Coronado[1],
Jorge Enrique Hernández González**[2]**, Dieter Willbold[1], Raphael Josef Eberle**[3]*

1 Institut für Physikalische Biologie, Heinrich-Heine-Universität Düsseldorf, Düsseldorf, Germany,
2 Institute of Biosciences, Humanities and Exact Sciences, São Paulo State University (UNESP), São José do Rio Preto, Brazil, 3 Institut für Biochemische Pflanzenphysiologie, Heinrich-Heine-Universität Düsseldorf, Düsseldorf, Germany

* eberler@hhu.de

## Abstract

Climate change has facilitated the spread of arboviruses like the Chikungunya virus (CHIKV). CHIKV, a re-emerging virus from the *Togaviridae* family, has caused numerous global outbreaks. The absence of antiviral therapy against CHIKV poses a significant threat to public health. The cleavage of the viral polyprotein relies on the catalytic activity of nsP2, crucial for viral replication. Therefore the nsP2 protease presents a promising target for antiviral drug development. Animal venom-derived peptides demonstrated potential in combating various diseases including infections, cancer, and neurodegenerative disorders. In this study, we assessed the inhibitory effects of pantinin-1, a peptide derived from the scorpion *Pandinus imperator* with broad antimicrobial activity, against CHIKV nsP2 protease. Pantinin-1 effectively inhibited CHIKV nsP2 protease, with a half-maximal inhibitory concentration ($IC_{50}$) of $6.4 \pm 2.04$ μM and complete inhibition at 175 μM. Further examination revealed that pantinin-1 functions as a competitive inhibitor with low micromolar affinity and exhibited no toxicity up to 20 μM in cell culture. Using docking and molecular dynamics simulations, the protein-peptide interaction was analyzed, and the key residues involved in the protease binding were predicted. These findings underscore the potential of pantinin-1 as a lead candidate targeting nsP2 protease.

## 1. Introduction

Arthropod-borne viruses (arboviruses) are a significant global health threat due to their ability to cause severe illness and widespread outbreaks [1]. These viruses are primarily transmitted by arthropods such as mosquitoes and ticks and include human pathogens such as dengue virus (DENV), Zika virus (ZIKV), and chikungunya virus (CHIKV), all of which are transmitted by Aedes mosquito species [2]. Among these viruses, CHIKV stands out due to its rapid geographic expansion, with cases reported in more than 100 countries across Africa, Asia, the Americas, and Europe

**Data availability statement:** All relevant data are within the manuscript and its Supporting information files.

**Funding:** Dr. González, CNPq, 309940/2019-2 and 403193/2022-2; FAPESP, 2024/13327-2, 2020/08615-8 and 2024/01956-5. Dr. Mastalipour had a PhD grant from the Jürgen Manchot Stiftung. The funders had no role in study design, data collection and analysis, decision to publish, or preparation of the manuscript.

**Competing interests:** The authors have declared that no competing interests exist.

[3–5]. CHIKV infection causes chikungunya fever, which typically presents in two clinical phases [6]. The acute phase is characterized by high fever, headache, rash, and severe joint pain [7]. In some patients, symptoms persist into a chronic phase, characterized by ongoing arthralgia, sleep disturbances, and psychological conditions such as depression [8,9]. Additionally, CHIKV has been associated with neurological complications, including encephalitis and peripheral neuropathies, particularly in neonates [10].

CHIKV is a single-stranded, positive-sense RNA virus belonging to the Togaviridae family, genus Alphavirus [11]. Its genome encodes four non-structural proteins (nsP1–nsP4) and five structural proteins including the capsid (C), envelope protein (E1,E2) and three accessory proteins (E3, transframe protein and 6k) [12,13]. Among these, non-structural protein 2 (nsP2) plays a key role in viral replication. It contains an RNA helicase domain with nucleoside triphosphatase (NTPase) and RNA triphosphatase activities, as well as a C-terminal protease and methyltransferase domain [14]. The nsP2 protease (nsP2$^{pro}$) is a member of MEROPS clan CN, features a papain-like cysteine protease domain and is responsible for cleaving the viral polyprotein P1234 into individual functional units essential for replication [15,16]. Due to its essential enzymatic functions, nsP2$^{pro}$ is a promising target for antiviral drug development. Several studies have explored potential inhibitors of nsP2$^{pro}$ using in silico, in vitro, and biochemical approaches. These include peptide-based inhibitors such as P1 and natural products like hesperidin [17,18]. However, none of these compounds have advanced beyond preclinical testing or received approval for human use. While two vaccines, VLA1553 [19] and Vimkunya [20], have recently been approved against CHIKV infection, vaccination is not effective for individuals who are already infected. This underscores the need for antiviral therapy.

In recent years, animal venoms from arthropods such as spiders and scorpions, as well as snakes and other venomous species have emerged as promising sources of bioactive compounds with pharmacological potential [21]. These venoms contain peptides and proteins with diverse mechanisms of action. They have been studied for applications including antimicrobial and antiviral therapies, as well as treatments for neurodegenerative diseases such as Alzheimer's and Parkinson's [22–25]. Besides the diversity and broad spectrum of venom-derived peptides, many of them exhibit advantages such as cell-penetrating properties, specificity, and resistance to enzymatic degradation [26,27]. For example, a peptide from Crotalus durissus terrificus venom has demonstrated in vitro activity against amyloid β$_{42}$, a key pathological hallmark of Alzheimer's disease [24]. In the context of infectious diseases, venom-derived peptides have shown broad-spectrum activity. Latarcins, derived from the spider Lachesana tarabaevi, have exhibited potent antibacterial and antiviral activity [28,29]. Similarly, the scorpion-derived peptide Mucroporin-M1 has shown virucidal activity against pathogens including SARS-CoV, measles virus, and influenza H5N1 [30]. Additional venom-derived compounds with antimicrobial or antiviral activity are summarized in Table 1. In addition to these known peptides, pantinin-1 has gained attention due to its potent and broad-spectrum antimicrobial properties [31]. Pantinin-1, a 14-amino acid peptide derived from the scorpion Pandinus imperator

**Table 1. Representative proteins isolated from venom and venom-derived peptides with reported antimicrobial, antiviral, or therapeutic activity.**

| Agent(s) | Species | Target | Reference |
|---|---|---|---|
| Captopril* | *Bothrops jararaca* | Hypertension | [33,34] |
| Contortrostatin | *Agkistrodon contortrix* | Breast cancer | [35] |
| Crotamine derivative peptides (CDP) Phospholipase A2 | *Crotalus durissus terrificus* | amyloid-$\beta_{42}$ (Alzheimer's disease), SARS-CoV2 Dengue virus and Yellow fever virus | [24,36] [37] |
| Echinhibin-1 | *Echis coloratus* | Sendai virus | [38] |
| Hemocoagulase | *Bothrops atrox* | Blood coagulation | [39] |
| Integrilin* | *Sistrurus miliarus barbourin* | Acute coronary syndrome | [34] |
| Latarcin | *Lachesana tarabaevi* | Gram-positive and Gram-negative bacteria, antiviral activity against Dengue virus | [28,29] |
| Mucroporin-M | *Lychas mucronatus* | SARS-CoV, influenza H5N1, Measles | [30] |

* FDA Approved.

[31,32], possesses a characteristic α-helical, amphipathic structure believed to underlie its membrane-disrupting mechanism and hemolytic activity [31]. It has demonstrated antimicrobial activity against Gram-positive bacteria such as Staphylococcus aureus (AB 94004) and MRSA (16472), Gram-negative strains such as Escherichia coli (DH5α) and Klebsiella oxytoca (AB 2010143), as well as antifungal activity against Candida tropicalis (AY 91009), with a minimum inhibitory concentration (MIC) of 16 μM [31].

In this study, we investigated the inhibitory potential of pantinin-1 against the CHIKV nsP2 protease *in vitro*. Our findings revealed that pantinin-1 effectively inhibited nsP2$^{pro}$ activity, with an IC$_{50}$ of 6.4 ± 2.04 μM. Further analysis showed that pantinin-1 binds at the active site as a competitive inhibitor with an equilibrium dissociation constant (K$_D$) of 9.29 μM. Cytotoxicity assessment using the MTT assay showed no detectable toxicity at concentrations up to 20 μM in HEK 293 and Vero cells, while approximately 50% cell viability was observed in Vero cells and around 70% in HEK 293 cells at 40 μM. *In silico* studies predicted the most stable pose and interaction between the protease and pantinin-1, identifying key residues involved in binding. While nsP2 protease inhibitors have been previously described, pantinin-1 advances the field by introducing a venom-derived peptide from an unexplored chemical space. Its compact size, defined binding mode, and inhibitory activity within a non-cytotoxic concentration range, position pantinin-1 as a promising starting point for further antiviral optimization.

## 2. Materials and methods

### 2.1. Protein expression and purification

The recombinant CHIKV nsP2 protease was expressed and isolated following a previously established and published protocol [17,18]. The purity of CHIKV nsP2$^{pro}$ samples is shown in the Supplementary Material (S1 Fig).

### 2.2. Peptide material

The peptide used in this study, pantinin-1, was provided in solid, lyophilized form by GenScript Biotech (Rijswijk, Netherlands). It consists of 14 L-amino acids (GILGKLWEGFKSIV) and was chemically modified with an acetyl group at the N-terminus and an amide group at the C-terminus. Analytical documentation provided by the supplier confirmed that the peptide met a purity threshold exceeding 90%. Verification of peptide identity and purity was conducted using chromatographic and spectrometric methods. Reverse-phase HPLC was performed with an Inertsil ODS-0310 analytical column (dimensions: 4.6 mm × 250 mm), and molecular weight confirmation was achieved via electrospray ionization mass

spectrometry (ESI-MS). The full analytical documentation supporting peptide purity and identity is included in the Supplementary Material (S2 Fig).

## 2.3. Enzymatic inhibition assay of CHIKV nsP2$^{pro}$

The inhibitory effect of pantinin-1 on CHIKV nsP2$^{pro}$ was evaluated using a fluorescence-based enzymatic assay. This assay utilized a synthetic peptide substrate labeled with DABCYL and EDANS fluorophores (DABCYL-Arg-Ala-Gly-Gly-↓Tyr-Ile-Phe-Ser-EDANS; BACHEM, Bubendorf, Swit-zerland), representing the native cleavage sequence of the CHIKV polyprotein [18,40]. The assay was conducted in a 96-well microplate with a total volume of 100 µL per well. Each reaction mixture contained 20 mM Bis-Tris propane buffer (pH 7.5), 10 µM nsP2$^{pro}$, and 9 µM fluorogenic substrate. Pantinin-1 was prepared as a 10 mM stock solution in DMSO and diluted into the reaction mixture to achieve final concentrations ranging from 0 to 175 µM. Depending on the stocksolution concentration the overall DMSO concentration did not exceed 2%. A DMSO control experiment assessing the effect on protease activity is depicted in the Supplementary Material (S3 Fig). 2% DMSO did not exhibit a significant impact on protease activity.

Fluorescence emission, reflecting substrate cleavage, was monitored at 37 °C using a CLARIOstar plate reader (BMG Labtech, Ortenberg, Ger-many) with excitation and emission wavelengths set to 340 nm and 490 nm, respectively. Readings were taken at 30-second intervals for 30 minutes. Enzymatic activity in the presence of pantinin-1 was expressed as a percentage of the activity observed in untreated control samples, calculated using the equation (E1) [18]:

$$\text{Relative protease activity \%} = \frac{\text{Intensity of enzymatic activity after inhibiton}}{\text{Intensity of enzymatic activity}} \times 100$$

To determine the IC$_{50}$ value, non-linear regression analysis was performed using GraphPad Prism (version 5). All experiments were performed as triplicates, and the results are presented as mean ± standard deviation (SD).

## 2.4. Inhibition mechanism analysis

To investigate how pantinin-1 interacts with the catalytic function of CHIKV nsP2$^{pro}$, an inhibition assay was performed using the same FRET-based platform previously described. The enzyme concentration was maintained at 10 µM, while levels of both substrate and inhibitor were systematically adjusted. All reactions were carried out in 20 mM Bis-Tris-Propane buffer, pH 7.5. The data obtained, which involved various combinations of substrate and inhibitor concentrations (Table 2), were utilized to analyze the inhibition pattern using global nonlinear fits of initial velocity data and Lineweaver–Burk plots.

## 2.5. Biolayer interferometry (BLI)

Biolayer interferometry (BLI) was utilized to determine the equilibrium dissociation constant (K$_D$) of pantinin-1. Experiments were performed using an Octet RED96 system (Sartorius, Göttingen, Germany) with Octet AR2G biosensors (Sartorius, Göttingen, Germany). The biosensors were incubated in running buffer (20 mM Bis-Tris propane, 2%

**Table 2. Substrate and peptide concentrations used in the inhibition-mode assay.**

| Substrate conc. [µM] | 0 | 1 | 2.5 | 5 | 7.5 | 10 |
|---|---|---|---|---|---|---|
| Inhibitor conc. [µM] | 0 | 0 | 0 | 0 | 0 | 0 |
| | 1 | 1 | 1 | 1 | 1 | 1 |
| | 2.5 | 2.5 | 2.5 | 2.5 | 2.5 | 2.5 |
| | 5 | 5 | 5 | 5 | 5 | 5 |

DMSO, pH 7.5) for 15 minutes before commencing the procedure. The experiments began with an equilibration step in running buffer for 600 seconds. Subsequently, the sensor surface was activated with an activation buffer containing 100 mM NHS and 50 mM MES, pH 5.2 (Xantec, Düsseldorf, Germany), for 420 seconds. The nsP2pro (10 µg/mL) was then immobilized onto the activated biosensor surface for 420 seconds in 20 mM sodium acetate, pH 5.0. The reaction was then quenched using 1 M ethanolamine-HCl, pH 8.5 (Xantec, Düsseldorf, Germany), for 300 seconds. Following this, sensors were re-equilibrated in running buffer for an additional 600 seconds. A six-step 1:3 serial dilution of pantinin-1 was prepared, ranging from 20 µM to 0 µM. Each sensor was exposed to pantinin-1 during an association step for 180 seconds, followed by a 600 seconds dissociation phase. After each cycle, sensors were regenerated with 20 mM glycine (pH 2.0) and re-equilibrated in running buffer for 600 seconds. All experiments were performed in triplicate. To investigate the binding of pantinin-1 on sensors, a control experiment was conducted as described above without immobilizing the protease, with the highest concentrations of pantinin-1 at 20 µM and 6.66 µM. Data were analyzed using ForteBio Data Analysis Software 8.0 (Sartorius, Göttingen, Germany). Binding responses from three independent experiments were combined to construct a Scatchard plot, from which the dissociation constant ($K_D$) was determined [41].

## 2.6. Cell viability assay

The cytotoxic potential of pantinin-1 was assessed in Vero cells (African green monkey kidney epithelial cell line, *Chlorocebus aethiops*) and HEK 293 (Human Embryonic Kidney) cells. The Vero cell line was generously provided by Dr. Sabrina Bergkamp from the Ernst Ruska-Centre for Microscopy and Spectroscopy with Electrons (ER-C), Structural Biology (ER-C-3), Forschungszentrum Jülich. Vero cells were cultured in Dulbecco's Modified Eagle Medium (DMEM) supplemented with 10% fetal calf serum (FCS) and 1% non-essential amino acids, and maintained at 37 °C in a humidified incubator with 5% $CO_2$. HEK 293 cells were cultured under the same conditions, except without the addition of non-essential amino acids. For the assay, cells were seeded in 96-well plates and treated with pantinin-1 at final concentrations ranging from 0 to 100 µM. The peptide was dissolved in DMSO with a final concentration of 10 mM and subsequently diluted in complete culture medium. To establish a control for maximal cytotoxicity, 0.1% Triton X-100 was used as a positive control. After overnight incubation with the test compound, cell viability was assessed using the MTT assay (Roche Diagnostics GmbH, Mannheim, Germany). 10 µl of MTT reagent was added to each well, followed by a 4 hour incubation to allow formazan formation by metabolically active cells. Crystals were dissolved by the addition of 100 µL of solubilization solution, and the plates were incubated overnight to ensure complete dis-solution. Absorbance was recorded at 570 nm and 660 nm using a CLARIOstar plate reader (BMG Labtech, Ortenberg, Germany). Cell viability was calculated using the following equation (E2):

$$\frac{(A570 - A660) \text{of treated cells}}{(A570 - A660) \text{of control (not treated cells)}} \times 100$$

## 2.7. Protein-peptide docking

An MD-derived nsP2pro conformation with an open active site, as reported in a previous work [17], was employed here to dock the pantinin-1 peptide. The three-dimensional structure of pantinin-1 used for docking was predicted using Alpha-Fold3 [42]. Three docking programs, Galaxy TongDock [43], ClusPro [44] and HDock [45], were used for this purpose with the default parameters described in their respective web servers. The best poses along with three poses showing high conformational variation (i.e., large pair-wise peptide RMSD values) among the top-5 solutions determined by each docking program were selected for subsequent analyses to predict the most likely binding mode of pantinin-1 to nsP2pro, as described below.

## 2.8. MD simulations

The selected nsP2$^{pro}$-pantinin-1 complexes were prepared for MD simulations with tleap of Amber22 [46] as described in Mastalipour et al. [17]. Briefly, the peptide was capped at the N- and C-termini with ACE and NME moieties. The parameters for both the peptide and the protein were drawn from the ff19SB force field [47]. Octahedral boxes were defined to simulate each system, with edges placed at least 10 Å away from the solute surface. The simulation boxes were then filled with OPC water molecules [48] and 16 Cl- counterions were added to neutralize the solute net charge. Each system underwent energy minimization, followed by NVT heating and equilibration at the NPT ensemble to ensure a final temperature and pressure of 300 K and 1 bar, respectively. Then, the stepwise decrease of the harmonic restraints applied to the solute's heavy atoms during the equilibration phase was conducted during four NPT simulations. Finally, 1 μs productive runs were run for every complex. All the MD simulations were performed with pmemd.cuda of Amber22 [46]. More details on the MD simulation setup can be found elsewhere [17].

## 2.9. MM-GBSA free energy calculations

Molecular Mechanics Generalized-Born Surface Area (MM-GBSA) calculations were performed for the three top-ranked poses of peptide P1 in complex with CHIKV nsP2$^{pro}$ using the MMPBSA.py module of Amber22 [46,49], following the protocol detailed in Mastalipour et al., 2025 [17]. The single-trajectory approach was employed, extracting 542 frames from the final 0.5 μs of each MD trajectory. The GB-neck2 model (igb = 8) [50] was used to estimate polar solvation energies, with internal and external dielectric constants set to 1 and 80, respectively, and a salt concentration of 0.1 M. Per-residue free energy decomposition was also performed under the same conditions to identify key binding interface residues [17].

## 2.10. Trajectory analyses

Trajectory analyses were conducted using the cpptraj module from Amber22 [46]. Root-mean-square deviation (RMSD) values were obtained via the rms command. Clustering analysis employed the cluster command with a hierarchical agglomerative algorithm, using RMSD of peptide heavy atoms as the distance metric. Typically, five clusters were generated, and the most populated cluster was selected for structural representation. Hydrogen bond analysis was performed with the hbond command, applying geometric criteria of a donor–hydrogen–acceptor angle greater than 120° and a hydrogen–acceptor distance of 3.2 Å or less to define intermolecular hydrogen bonds.

## 2.11. Statistical analysis

All statistical analyses were conducted using GraphPad Prism version 5.0. Differences between treatment groups and the control were evaluated using one-way analysis of variance (ANOVA) followed by Tukey's post hoc test for multiple comparisons. Statistical significance was denoted as follows: $p < 0.05$ (*), $p < 0.01$ (**), and $p < 0.001$ (***).

## 3. Result

### 3.1. Inhibitory effect of pantinin-1 on CHIKV nsP2$^{pro}$ activity

CHIKV nsP2$^{pro}$ was expressed and purified as previously described by Eberle et al., 2021 and Mastalipour et al., 2025 [17,18]. The pure protease was used to test the inhibitory potential of the α-helical, amphipathic peptide pantinin-1 (Fig 1A) against the proteolytic activity of CHIKV nsP2$^{pro}$. This activity was assessed using a FRET-based enzymatic assay. Enzymatic activity was determined by measuring fluorescence emission over a 30-minute period at excitation and emission wavelengths of 340 and 490 nm, respectively. As shown in Fig 1B, pantinin-1 exhibited a clear, concentration-dependent inhibitory effect on nsP2$^{pro}$. A reduction in enzymatic activity was observed at low micromolar concentrations, with approximately 50% inhibition at around 5 μM. This indicates that pantinin-1 can efficiently interfere with nsP2$^{pro}$

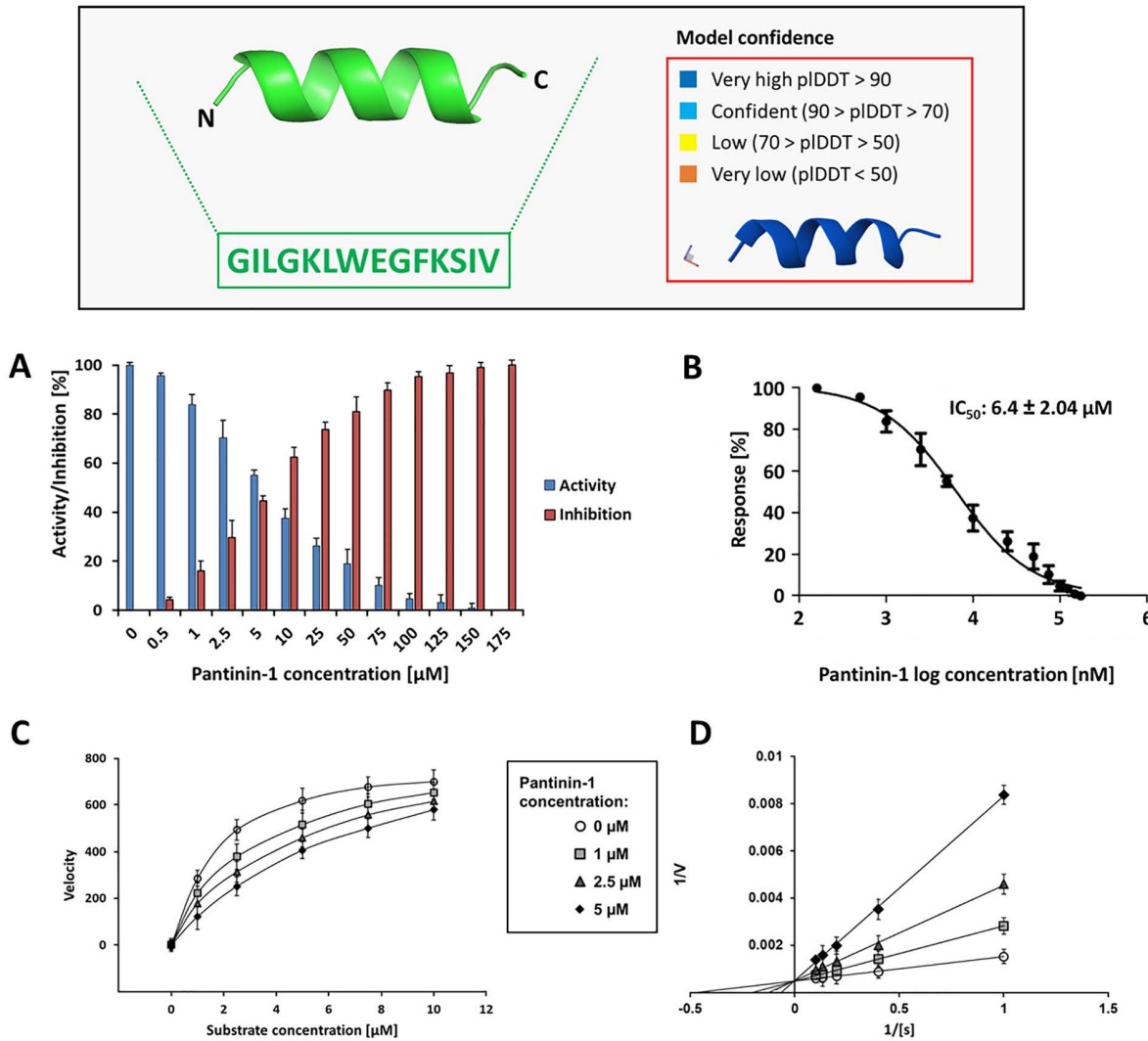

**Fig 1. Pantinin-1 structure prediction and its inhibitory activity and binding analysis of pantinin-1 toward CHIKV nsP2^pro. (Box)** The box in the upper panel of the figure shows the predicted structure of pantinin-1 generated by AlphaFold3 and its sequence. The predicted structure is shown in ribbon view. The AlphaFold3 confidence score of the pantinin-1 model is > 90%. The following experiments are shown as mean ± SD (n = 3 technical replicates). **(A)** Concentration-dependent inhibition of CHIKV nsP2^pro activity by pantinin-1, measured using a fluorescence-based enzymatic assay. Fluorescence was recorded over 30 min at 340/490 nm and 37 °C. Enzymatic activity was expressed as a percentage relative to the untreated control. **(B)** Dose–response curve of pantinin-1 inhibition, generated by non-linear regression analysis using GraphPad Prism. The calculated $IC_{50}$ value was 6.4 ± 2.04 μM. **(C)** Global nonlinear fits of initial velocity data were used to determine the inhibition mode of Pantinin-1. The data were transformed into Lineweaver–Burk plots (1/V0 versus 1/[S]) **(D)**. Both analysis showed an increase in the Michaelis constant ($K_m$) and no effect on the maximum velocity ($V_{max}$), which is characteristic of a competitive inhibitor.

activity at relatively low concentrations. The inhibitory effect increased with higher peptide concentrations, leading to a complete loss of enzymatic activity at 175 μM, the highest concentration tested. To quantify the potency of pantinin-1, a dose-response curve was generated, and the $IC_{50}$ value was calculated using non-linear regression analysis in GraphPad. As shown in Fig 1C, the $IC_{50}$ was determined to be 6.4 ± 2.04 μM, based on three independent replicates.

## 3.2. Mode of inhibition of pantinin-1 toward CHIKV nsP2$^{pro}$

In Section 3.1, we demonstrated the inhibitory effect of pantinin-1 on the enzymatic activity of CHIKV nsP2$^{pro}$. To further characterize this interaction, it was essential to investigate the mechanism and mode of inhibition of pantinin-1. To achieve this, a FRET-based enzymatic assay was conducted using varying concentrations of both substrate and pantinin-1. The initial velocity data were fitted using global nonlinear approach (Fig 1C) and then transformed into a Lineweaver–Burk plot (1/V0 versus 1/[S]) (Fig 1D). Increasing concentrations of Pantinin-1 shifted the dose-response curve to the right in the nonlinear fitting (Fig 1C) and in the Linewaever-Burk plot it led to an increase in the slopes of the linearized data. These changes occurred without altering the y-axis intercept (1/V$_{max}$) but by varying, the x-axis intercepts (−1/K$_m$) (Fig 1D). Both analyses demonstrated that increasing the concentration of Pantinin-1 at multiple fixed substrate concentrations resulted in a progressive increase in the apparent K$_m$ constant. This pattern indicates that pantinin-1 decreases the substrate's binding affinity to nsP2$^{pro}$.

## 3.3. Equilibrium dissociation constant measurement by biolayer interferometry (BLI)

To determine the equilibrium dissociation constant (K$_D$) for the interaction between pantinin-1 and CHIKV nsP2$^{pro}$, bio-layer interferometry (BLI) was used. The experiment was performed using an Octet RED96 system (Sartorius, Göttingen, Germany) equipped with AR2G biosensors. Pantinin-1 was tested at six different concentrations, ranging from 20 µM to 0 µM. A sensor exposed only to the running buffer during the assay served as the reference (0µM). All experiments were conducted at room temperature. The assay included an association phase of 180 s followed by a dissociation phase of 600 s. A representative sensorgram is shown in S4A Fig. All measurements were conducted in triplicate. From the BLI binding response data obtained across six concentrations of pantinin-1, a Scatchard plot was constructed using values extracted from ForteBio Data Analysis Software. This analysis resulted in a dissociation constant (K$_D$) of 9.29 µM (S4B Fig). To confirm that the observed binding was specifically between the protease and pantinin-1, a control experiment was performed in the absence of the protease using 20 µM and 6.66 µM pantinin-1. As shown in S5 Fig, no binding was detected between the sensor and the peptide, indicating that the interaction occurs exclusively between the protease and pantinin-1.

## 3.4. Cytotoxicity assessment of pantinin-1

To investigate the cytotoxic potential of pantinin-1, Vero and HEK 293 cells were exposed to a concentration gradient (0–100 µM) of the peptide, which was initially dissolved in DMSO and subsequently diluted in cell culture medium. Cell viability was determined using the MTT assay. As illustrated in Fig 2, pantinin-1 did not exhibit any detectable cytotoxicity at concentrations up to 20 µM in both cell lines, with viability levels similar to those of the untreated control. A significant decrease in viability was observed at 40 µM, where approximately half of the Vero cell population remained viable, and around 70% viability was observed in HEK 293 cells. Exposure to higher concentrations (60–100 µM) led to a progressive and substantial loss of cell viability in both cell lines, with HEK 293 cells showing around 50% viability and Vero cells dropping below 20% at 60 µM. High cytotoxicity was observed at 80 µM and above.

## 3.5. Prediction of the nsP2$^{pro}$-pantinin-1 complex structure

Protein-peptide docking was performed to predict the binding mode of pantinin-1 to nsP2$^{pro}$. The top-5 poses determined with the three docking algorithms chosen in this work are shown in Fig 3. Most of the solutions point at the CHIKV nsP-2$^{pro}$ active site as the region preferentially targeted by pantinin-1. However, ClusPro and HDock also predicted alternative binding modes, with pantinin-1 interacting at the interdomain hinge region located opposite the active site (Fig 3B-C).

As described in the Materials and Methods section, the best pose predicted by each docking algorithm, along with three poses selected from the top-5 poses generated by each algorithm, were chosen for subsequent analyses to determine

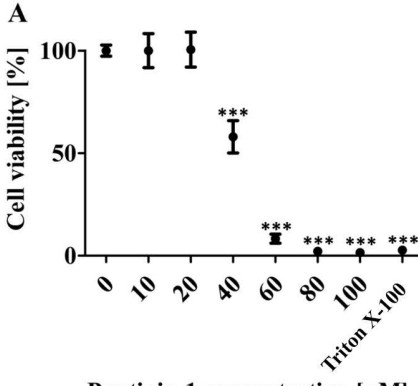 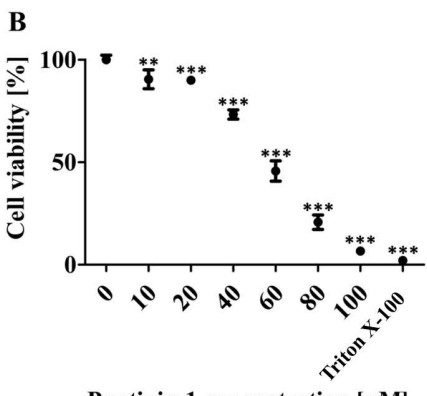

**Fig 2. Cytotoxicity of pantinin-1 in Vero and HEK 293 cells assessed by MTT assay. (A)** Vero cells and **(B)** HEK 293 cells were exposed to pantinin-1 at concentrations ranging from 0 to 100 µM. The peptide was dissolved in DMSO and diluted in culture medium. Cell viability was assessed using the MTT assay. In both cell lines, no cytotoxic effects were observed up to 20 µM. Viability decreased to approximately 50% at 40 µM in Vero cells and to about 70% in HEK 293 cells. At higher concentrations (80–100 µM), cell viability declined significantly, with most cells non-viable at 100 µM. To establish a control for maximal cytotoxicity 0.1% Triton X-100 was used. Data represent mean ± SD (n = 3 technical replicates). Statistical significance was determined by one-way ANOVA (**p < 0.01, ***p < 0.001 vs. untreated control).

their stability (S1 Table). The peptide RMSD values calculated during the MD simulations of the selected docking poses indicate that, in at least three systems (Galaxy TongDock pose3, and ClusPro poses 0 and 4), the peptide reached stable conformations, characterized by RMSD plateau regions (Fig 3). Moreover, the calculated $\Delta G_{eff}$ values for all simulated systems predict that the conformation adopted during the second half of the MD simulation initiated from ClusPro pose 4 is the most stable.

The representative structure calculated for the trajectory of the most stable pose in complex with CHIKV nsP2pro (ClusPro pose 4, Fig 3B), is depicted in Fig 4A. As can be observed, the final pantinin-1 binding mode diverges significantly from the docking pose, as also deduced by the peptide RMSD profile (Figs 3B and 4A). At the new position, the peptide occupies an active site pocket at the interdomain region, while leaving the catalytic residues exposed. However, the predicted binding mode is still capable of sterically hindering substrate access to the active site. This is supported by the superposition of nsP2pro in complex with pantinin-1 and the crystal structure of the Venezuelan equine encephalitis virus (VEEV) nsP2pro containing an N-terminal segment in a substrate-like conformation within the active site (S6 Fig).

The free energy decomposition analysis suggests a dominant contribution of hydrophobic interactions, as deduced from the nature of the most important residues at the interface (Fig 4B). Residues L3, W7, F10, I13 and V14 of pantinin-1 are predicted as key residues for nsP2pro binding. Their mutation to Ala can serve as a strategy to experimentally validate the conclusions derived from the in silico analyses presented here.

## 4. Discussion

Chikungunya virus (CHIKV), along with Dengue, Yellow Fever, and Zika viruses, is among the major emerging arboviruses transmitted by Aedes species mosquitoes and is responsible for outbreaks worldwide [1,51]. CHIKV belongs to the genus Alphavirus [2], and its replication depends on the multifunctional non-structural protein 2 (nsP2). The C-terminal region of nsP2 contains a protease domain (nsP2pro), which plays a crucial role in processing the viral polyprotein, an essential step in the viral life cycle [16]. Over the years, various strategies have been explored to inhibit CHIKV nsP2pro as a potential antiviral target. Techniques such as phage display have been used to identify inhibitory compounds [17]. Although numerous candidates have been reported, none have advanced beyond the laboratory stage (Table 3).

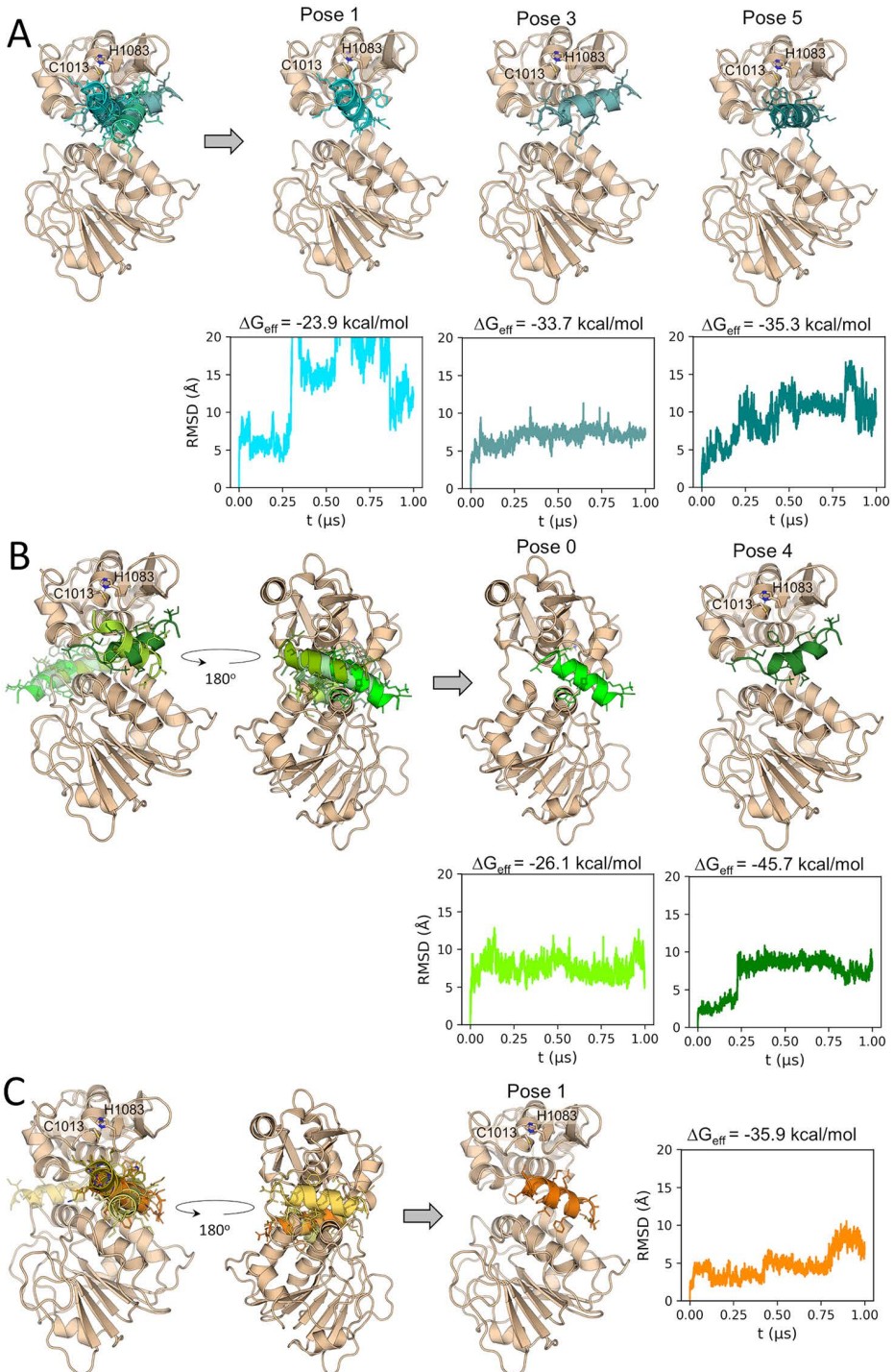

**Fig 3. Prediction of the CHIKV nsP2$^{pro}$-pantinin-1 complex through docking, MD simulations and free energy calculations.** Superimposed five top-scoring docking poses of pantinin-1 predicted with **(A)** Galaxy TongDock, **(B)** ClusPro and **(C)** HDock. Docking poses selected for MD simulations are depicted after the gray arrows. The pantinin-1 RMSD time profiles along the MD simulations of the predicted complexes are shown using the same colors as the corresponding peptide poses. The effective binding free energies ($\Delta G_{eff}$) calculated from the MD simulations are shown above each RMSD graph. RMSD values were calculated for the peptide backbone atoms and with respect to the peptide's conformation in the docking pose. Prior to peptide RMSD calculations, all trajectory frames were superimposed on the nsP2$^{pro}$ backbone in the initial conformation. The catalytic residues C1013 and H1083 are represented as sticks and labeled in the structural representation of the different CHIKV nsP2$^{pro}$-pantinin-1 docking poses.

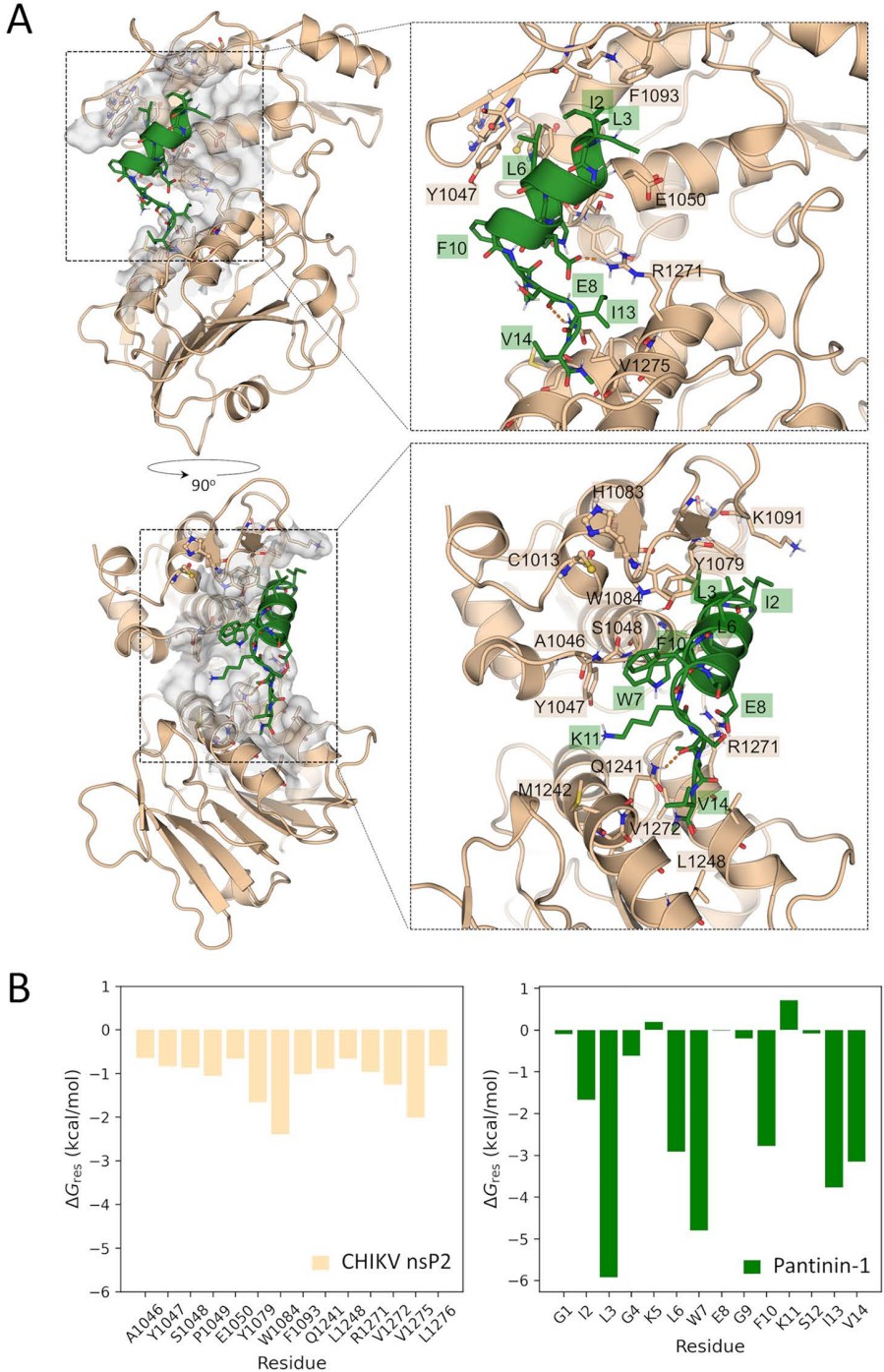

**Fig 4. Representative structure of the predicted CHIK nsP2$^{pro}$:pantinin-1 complex and the energy contribution of residues at the interface. (A)** Two views of the CHIKV nsP2$^{pro}$:pantinin-1 complex. Interface residues are shown in sticks and are labeled. The active site residues C1013 and H1083 are represented as sticks and spheres. H-bonds are indicated as orange dashed lines**. (B)** Per-residue free energy contributions of the nsP2$^{pro}$ interface residues (<0.6 kcal/mol) and of all pantinin-1 residues.

**Table 3. Summary of characterized CHIKV nsP2^pro inhibitors from prior research.**

| Compound | Type | IC$_{50}$ (µM) | K$_D$ (µM) | Inhibition Mode | Method IC$_{50}$/K$_D$ | Reference |
|---|---|---|---|---|---|---|
| Peptide P1 | Peptide | 4.6±1.9 | 1.4±0.6 | Competitive | FRET/MST | [17] |
| RA-0002034 | Small molecule | 58±17 | N/A | Covalent (Cys- binding) | Cell based/- | [60] |
| Withaferin A (WFA) | Natural compound | 0.51 (In BHK-21 cells) | 64 | N/A | FRET/MST | [67] |
| Hesperetin (HST) | Natural flavonoid | 2.5±0.4 | 31.6±2.5 | Noncompetitive | FRET/ fluorometry | [18] |
| Hesperidin (HSD) | Natural flavonoid | 7.1±1.1 | 40.7±2.0 | Noncompetitive | FRET/ fluorometry | [18] |
| Pep-I | Peptoid | 34 | N/A | Noncompetitive | FRET/- | [68] |
| Pep-II | Peptoid | 42 | N/A | Competitive | FRET/- | [68] |
| MBZM-N-IBT | Small Molecule | 32 | 6.1±0.6 | Competitive | FRET/ fluorometry | [69] |

Venom-derived peptides present a novel approach to develop a new therapeutic agent. These peptides have shown potential in treating a wide range of diseases, including bacterial and viral infections, cancer, and neurodegenerative disorders. For instance, Phospholipase A2, which is isolated from the venom of Crotalus durissus terrificus, has shown antiviral activity against Dengue and Yellow Fever viruses [37]. Ziconotide, a peptide from the marine snail Conus magus, is approved for chronic pain management [52,53]. Mastoparan and its synthetic analogs, derived from wasp venom, exhibit antiviral activity against Human alphaherpesvirus 1 (HSV-1) [54]. Similarly, Dermaseptins from hylid frogs have demonstrated broad-spectrum antimicrobial and anti-tumor properties [55]. Pantinin-1 and −2, isolated from the scorpion Pandinus imperator, are known to have activity against Gram-positive bacteria, fungi, and viruses. Besides this broad antimicrobial impact, it was shown that pantinin-1 displays substantial stability in human serum and a strong resistance to proteolytic degradation. After 16 hours of incubation, approximately 70–80% of the peptide remained intact. Furthermore, pantinin-1 caused no significant hemolysis of human erythrocytes at concentrations up to 512 µg/mL. While these results indicate the potential for an extended biological half-life, detailed in vivo pharmacokinetic data have yet to be reported [56–59].

In this study, we evaluated pantinin-1 as a potential inhibitor of CHIKV nsP2^pro. A FRET-based assay was conducted using pantinin-1 at concentrations ranging from 0 to 175 µM to determine the inhibitory effect of pantinin-1. As shown in Fig 1B, increasing concentrations of pantinin-1 progressively inhibited protease activity, with complete inhibition observed at 175 µM. Dose–response analysis (Fig 1C) revealed an IC$_{50}$ of 6.4±2.04 µM. This potency is comparable to that of other known inhibitors (Table 3) such as Peptide P1 (4.6 µM) [17], Hesperetin (2.5 µM) [18], and Hesperidin [18], although it is higher than that of RA-0002034, which exhibited a much lower IC$_{50}$ of 58±17 nM [60]. To understand the mechanism of inhibition, we used global nonlinear fitting of initial velocity data and Lineweaver–Burk analysis using varying concentrations of both substrate and pantinin-1. The resulting plots (Fig 1C and D) showed increased K$_m$ and unchanged V$_{max}$ values indicating that pantinin-1 acts as a competitive inhibitor [61]. Pantinin-1 alters the K$_m$ value of CHIKV nsP2^pro in a manner that, in both global nonlinear fitting and Lineweaver–Burk plot representations, resembles the effects observed for known competitive inhibitors of several enzymes and proteins, including hemolysin, human cathepsin L, enzyme I, phosphate transporter and cytochrome P450 [62–66]. This mode of inhibition is consistent with what we previously observed for peptide P1 [17]. Notably, both pantinin-1 and peptide P1 share a high content of α-helical secondary structure, which we have previously suggested may play a key role in peptide binding to the nsP2^pro active site. Next, we evaluated the binding affinity of pantinin-1 to CHIKV nsP2^pro using biolayer interferometry (BLI). Binding responses were measured across a concentration range of 0–20 µM, and values extracted from three independent experiments were used to construct a Scatchard plot. As shown in S4B Fig, this analysis yielded a dissociation constant (K$_D$) of 9.29 µM. While the binding affinity of pantinin-1 is slightly weaker than that of peptide P1 (1.39 µM) [17], it is significantly stronger than that of

Hesperetin ($31.6 \pm 2.5\mu M$) and Hesperidin ($40.7 \pm 2.0$ µM), as reported by Eberle et al. [18]. This difference does not necessarily indicate that pantinin-1 has superior binding, as various methods were used to determine the $K_D$ of the inhibitors. For example, in the case of peptide P1, microscale thermophoresis was used, which underscores that this difference in affinity may lie in the different methods used.

We also assessed the cytotoxicity of pantinin-1 on Vero and HEK 293 cells across a concentration range of 0–100 µM. As shown in Fig 2, pantinin-1 exhibited no cytotoxic effects up to 20 µM in both cell lines. However, at 40 µM, cell viability dropped to about 50% in Vero cells and approximately 70% in HEK 293 cells. Between 80–100 µM, the majority of cells in both lines were non-viable. We observed a slight variation in cytotoxicity between the two cell lines, with HEK 293 cells demonstrating slightly higher tolerance to pantinin-1. Nevertheless, in both cell lines, cytotoxic effects began to appear at concentrations exceeding 20 µM. As previously mentioned, pantinin-1 is an α-helical, amphipathic peptide [31]. It exhibits both antimicrobial and hemolytic properties by interacting with and disrupting cell membranes [31]. This characteristic explains the toxicity observed in the MTT assay, where cell death was evident at higher concentrations of pantinin-1. In addition, Zeng et al. reported that the hemolytic activity of pantinin-1 initiates at concentrations above 32 µM, aligning with the cytotoxic effects observed in this study [31]. Even though cytotoxicity is only evident at concentrations above 20 µM, this still represents more than three times the $IC_{50}$.

During the molecular docking studies with three different programs (ClusPro [44], HDock [45] and Galaxy TongDock [43]), we predicted likely conformations for the protease/pantinin-1 complexes. Most of these complexes showed binding at the active site of the protease (Fig 3A). However, two of the programs exhibited conformations within the top-5 poses where pantinin-1 binds to a region opposite the active site (Fig 3B–C). Further analysis, which included MD simulations followed by MM-GBSA free energy calculations, led to the identification of the most stable binding mode of pantinin-1 to CHIKV nsP2$^{pro}$ (Pose 4; Fig 3B). In this conformation, pantinin-1 interacts with active site residues located in both domains of the protease (Fig 4A). The proposed binding mode aligns with the competitive inhibition mechanism determined for this peptide. Free energy decomposition analysis demonstrated that hydrophobic interactions are the dominant force between pantinin-1and CHIKV nsP2$^{pro}$. The lowest per-residue free energies ($\Delta G_{res}$) for pantinin-1 correspond to L3, W7, F10, I13, and V14, indicating their important role in the interaction. However, this finding should be validated, possibly through alanine substitution. Moreover, as shown in Fig 4B, CHIKV nsP2$^{pro}$ residues Y1079, Y1047, and W1084 are involved in the interaction with the peptide. These amino acids are part of the substrate-binding site subunits S2, S3, and S4, suggesting partial occupancy of the substrate-binding site [70].

Peptide-based antiviral leads often exhibit low micromolar inhibitory activity at early stages of development, frequently accompanied by a relatively narrow therapeutic window that necessitates subsequent optimization [71–73]. The amphipathic and α-helical nature of pantinin-1, while contributing to its bioactivity, may also underlie the observed cytotoxicity at higher concentrations, a characteristic commonly reported for membrane-active and venom-derived peptides [74–76]. Importantly, such limitations can be mitigated through peptide engineering strategies, including sequence truncation, charge redistribution, reduction of hydrophobicity, and incorporation of non-natural or D-amino acids to improve selectivity and safety profiles [77–79]. Given its short length and defined structure, pantinin-1 represents a suitable scaffold for rational optimization aimed at enhancing its therapeutic index. Similar optimization approaches have successfully advanced other peptide antivirals from micromolar leads to clinically relevant candidates [80,81].

Overall, we demonstrated that pantinin-1, in addition to its antimicrobial activity, exhibits an inhibitory effect against CHIKV nsP2$^{pro}$ *in vitro*, with binding affinity and an $IC_{50}$ in the low micromolar range. This makes pantinin-1 a promising candidate for antiviral therapy development.

## 5. Conclusion and outlook

In this study, it was demonstrated that pantinin-1, a peptide derived from scorpion, has the potential to inhibit CHIKV nsP2$^{pro}$ *in vitro* with an $IC_{50}$ of $6.4 \pm 2.04$ µM and a binding affinity in the low micromolar range. Molecular docking and MD

simulation analyses revealed that, in its most stable state, pantinin-1 likely binds to the substrate binding site. The inhibition mode analysis confirmed this result by showing that pantinin-1 acts as a competitive inhibitor. In addition, the cytotoxicity assay demonstrated that pantinin-1 exhibited no toxic effects on Vero or HEK 293 cells at concentrations up to 20 µM. The present work establishes pantinin-1 as an inhibitor of CHIKV nsP2$^{pro}$ and as an early stage lead peptide. After further optimization of the peptide regarding its potency, metabolic stability, cell uptake and overall pharmacological properties, it will enter the next investigation level, from *in vitro* to *in vivo* experiments. In this way, the determination of EC$_{50}$ values using plaque-reduction and viral RNA assays will be essential to confirm its potential as an antiviral lead.

## Supporting information

**S1 Fig. Representative SDS-PAGE gel of purified CHIKV nsP2$^{pro}$ after size exclusion chromatography (SEC).** M: Marker (Protein ruler with corresponding molecular weight in kDa). SN: Sample loaded on the SEC column. Elution: Eluted protein after SEC.
(TIF)

**S2 Fig. HPLC chromatograms and mass spectrometry data of pantinin-1. (A)** Mass spectrometry data provided by GenScript Biotech (Netherlands) confirming the molecular weight and purity of the synthesized Pantinin-1 peptide. **(B)** HPLC chromatogram showing the retention time and purity profile of pantinin-1.
(TIF)

**S3 Fig. Impact of DMSO on CHIKV nsP2$^{pro}$ protease activity.** A control experiment was performed to assess the influence of DMSO on protease activity at concentrations of up to 20%. No significant changes in protease activity were detected at DMSO concentrations ≤6%, indicating that DMSO is an appropriate solvent within this range. The rose box denotes the DMSO concentration range applied in the inhibition assays. Differences among groups were evaluated using one-way ANOVA with Tukey's multiple-comparison test. Significant differences relative to the control (0 µM inhibitor) are indicated by asterisks (***, $p < 0.001$). Values represent mean±SD ($n = 3$).
(TIF)

**S4 Fig. Representative BLI sensorgram and scatchard plot analysis of pantinin-1 binding to immobilized CHIKV nsP2$^{pro}$. (A)** Measurement performed using Octet AR2G biosensors. Pantinin-1 was tested at six concentrations (0–20 µM). The sensorgram includes a 180-second association phase followed by a 600-second dissociation phase at room temperature. **(B)** Scatchard plot of pantinin-1 binding to CHIKV nsP2$^{pro}$ based on BLI response data. Binding values at five concentrations were extracted from sensorgrams and plotted as Bound/Free versus Bound. The data are shown as mean±SD ($n = 3$ technical replicates). The dissociation constant ($K_D$) was calculated from the slope of the linear regression using the equation ($K_D$) = −1/slope [1] yielding a ($K_D$) of 9.29 µM.
(TIF)

**S5 Fig. Representative BLI sensorgram showing the control experiment for nonspecific binding of pantinin-1.** Measurements were performed using Octet AR2G biosensors without immobilized CHIKV nsP2$^{pro}$ to assess non-specific interaction with the sensor surface. Pantinin-1 was tested at two concentrations (20 µM and 6.66 µM). Each sensorgram includes a 180-second association phase and a 600-second dissociation phase at room temperature. No measurable binding response was detected at either concentration, indicating that pantinin-1 does not interact with the biosensor surface in the absence of the target protein.
(TIF)

**S6 Fig. Structural superposition of the predicted CHIKV nsP2$^{pro}$–pantinin-1 complex and the crystal structure of VEEV nsP2$^{pro}$.** The figure showing an N-terminal segment inserted into the active site in a substrate-like conformation (PDB ID: 8DUF).
(TIF)

**S1 Table. Top-5 docking poses for pantinin-1 predicted by three docking algorithms along with their backbone RMSD values relative to Galaxy TongDock pose 1.**
(DOCX)

## Acknowledgments

J.E.H.G. thanks the National Laboratory for Scientific Computing (LNCC/MCTI, Brazil) for providing HPC resources of the SDumont supercomputer, URL: http://sdumont.lncc.br. Figures were created using BioRender and GraphPad (version 5).

## Author contributions

**Conceptualization:** Mohammadamin Mastalipour, Raphael Josef Eberle.

**Data curation:** Mohammadamin Mastalipour, Jorge Enrique Hernández González.

**Formal analysis:** Mohammadamin Mastalipour, Mônika Aparecida Coronado, Jorge Enrique Hernández González, Raphael Josef Eberle.

**Investigation:** Mohammadamin Mastalipour.

**Methodology:** Mohammadamin Mastalipour, Jorge Enrique Hernández González.

**Resources:** Dieter Willbold.

**Supervision:** Raphael Josef Eberle.

**Writing – original draft:** Mohammadamin Mastalipour, Mônika Aparecida Coronado, Jorge Enrique Hernández González, Dieter Willbold, Raphael Josef Eberle.

**Writing – review & editing:** Mohammadamin Mastalipour, Mônika Aparecida Coronado, Jorge Enrique Hernández González, Dieter Willbold, Raphael Josef Eberle.

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
