## [Decision Letter · Decision Letter 0]

19 Nov 2025

PONE-D-25-43080Inhibition of Chikungunya virus nsP2 protease in vitro by scorpion venom peptide pantinin-1PLOS ONE

Dear Dr. Eberle,

Thank you for submitting your manuscript to PLOS ONE. After careful consideration, we feel that it has merit but does not fully meet PLOS ONE’s publication criteria as it currently stands. Therefore, we invite you to submit a revised version of the manuscript that addresses the points raised during the review process.

We look forward to receiving your revised manuscript.

Kind regards,

Seth Agyei Domfeh, PhD

Academic Editor

PLOS ONE

Additional Editor Comments (if provided):

Reviewers' comments:

Reviewer's Responses to Questions

**Comments to the Author**

1. Is the manuscript technically sound, and do the data support the conclusions?

Reviewer #1: Yes

Reviewer #2: Partly

2. Has the statistical analysis been performed appropriately and rigorously? 

Reviewer #1: Yes

Reviewer #2: Yes

3. Have the authors made all data underlying the findings in their manuscript fully available?

The PLOS Data policy requires authors to make all data underlying the findings described in their manuscript fully available without restriction, with rare exception (please refer to the Data Availability Statement in the manuscript PDF file). The data should be provided as part of the manuscript or its supporting information, or deposited to a public repository. For example, in addition to summary statistics, the data points behind means, medians and variance measures should be available. If there are restrictions on publicly sharing data—e.g. participant privacy or use of data from a third party—those must be specified.requires authors to make all data underlying the findings described in their manuscript fully available without restriction, with rare exception (please refer to the Data Availability Statement in the manuscript PDF file). The data should be provided as part of the manuscript or its supporting information, or deposited to a public repository. For example, in addition to summary statistics, the data points behind means, medians and variance measures should be available. If there are restrictions on publicly sharing data—e.g. participant privacy or use of data from a third party—those must be specified.requires authors to make all data underlying the findings described in their manuscript fully available without restriction, with rare exception (please refer to the Data Availability Statement in the manuscript PDF file). The data should be provided as part of the manuscript or its supporting information, or deposited to a public repository. For example, in addition to summary statistics, the data points behind means, medians and variance measures should be available. If there are restrictions on publicly sharing data—e.g. participant privacy or use of data from a third party—those must be specified.requires authors to make all data underlying the findings described in their manuscript fully available without restriction, with rare exception (please refer to the Data Availability Statement in the manuscript PDF file). The data should be provided as part of the manuscript or its supporting information, or deposited to a public repository. For example, in addition to summary statistics, the data points behind means, medians and variance measures should be available. If there are restrictions on publicly sharing data—e.g. participant privacy or use of data from a third party—those must be specified.

Reviewer #1: Yes

Reviewer #2: No

4. Is the manuscript presented in an intelligible fashion and written in standard English?

Reviewer #1: Yes

Reviewer #2: No

5. Review Comments to the Author

Reviewer #1: This study investigates the inhibitory activity of pantinin-1, a venom-derived antimicrobial peptide from Pandinus imperator, against the nonstructural protein 2 protease (nsP2pro) of Chikungunya virus (CHIKV), a key enzyme in viral replication. Using a FRET-based assay, pantinin-1 was shown to inhibit nsP2pro in vitro with an IC₅₀ of 6.4 μM. Kinetic analysis confirmed a competitive inhibition mechanism, while biolayer interferometry determined a dissociation constant (KD) of 9.29 μM. Cytotoxicity assays in HEK 293 and Vero cells revealed no toxicity up to 20 μM, with moderate effects at 40 μM and higher. In silico molecular docking and molecular dynamics simulations supported binding at the protease active site, identifying key hydrophobic residues (L3, W7, F10, I13, V14) and substrate-binding residues (Y1079, Y1047, W1084) as critical for interaction.

The results demonstrate that pantinin-1 exerts potent inhibitory activity against CHIKV nsP2pro within a concentration range below the cytotoxic threshold. These findings suggest that pantinin-1 may serve as a lead compound for developing antiviral agents against CHIKV. As the authors noted, further investigations, particularly in virus-infected cell models, are required to validate its antiviral activity. The experiments are properly designed and executed, and the conclusions are convincingly supported by the data. As a reviewer, I have no critical comments on the manuscript and recommend its publication in PLOS One .

Reviewer #2: The manuscript titled “Inhibition of Chikungunya virus nsP2 protease in vitro by scorpion venom peptide pantinin-1” investigates the antiviral potential of pantinin-1, a 14-amino-acid antimicrobial peptide derived from Pandinus imperator. Using a series of biochemical and biophysical assays, the authors demonstrate that pantinin-1 inhibits CHIKV nsP2 protease activity with an IC₅₀ of 6.4 ± 2.04 µM and shows competitive inhibition behaviour. Biolayer interferometry reveals a KD of ~9 µM, suggesting direct binding to nsP2pro. Cytotoxicity assays show no major toxicity up to 20 µM. Molecular docking and MD simulations identify a putative binding mode at the substrate-binding pocket. Overall, the manuscript presents a novel venom-derived peptide as a promising nsP2pro inhibitor and provides a mix of experimental and computational evidence supporting its potential as an antiviral lead.

Major Comments

1. The concept of targeting nsP2 protease is well established, and the authors correctly highlight the scarcity of potent inhibitors. Demonstrating inhibition by pantinin-1 is novel and relevant. However, the manuscript would benefit from a clearer articulation of how pantinin-1 advances the field beyond prior peptide inhibitors (e.g., P1 peptide, RA-0002034). At present, the novelty seems incremental.

2. A major limitation is the absence of direct antiviral assays (e.g., CHIKV infection in Vero/BHK cells). Since cytotoxicity data are available, inclusion of EC₅₀ values (or at least plaque-reduction and viral RNA quantification) would substantially strengthen the conclusions. Without this, it is difficult to evaluate whether the protease inhibition observed in vitro translates into antiviral activity.

3. The Lineweaver–Burk analysis suggests competitive inhibition, but these double-reciprocal plots are prone to error and are no longer recommended as standalone mechanistic proof. Consider supplementing with nonlinear global fitting of Michaelis–Menten data/enzyme progress curve analysis.

4. The MD simulations reveal stable poses, but the conclusion that pantinin-1 is a competitive inhibitor because it binds the active site is not fully convincing. The final MD pose appears shifted from the catalytic dyad, raising questions about how substrate exclusion occurs mechanistically. The authors should explicitly compare the peptide binding pose with the native substrate (P1234) cleavage motif or model the peptide and substrate simultaneously. Per-residue decomposition identifies strong hydrophobic contributions, but experimental validation (e.g., alanine-scanning) is essential before stating these as "key binding residues."

5. Although the peptide is non-toxic up to 20 µM, its IC50 is 6.4 µM, leaving a narrow therapeutic window. The discussion should compare this with the typical requirements for peptide antiviral drugs and address potential strategies to reduce cytotoxicity (e.g., peptide engineering, truncation, amino acid modification, etc.).

6. Protease resistance, serum stability, haemolytic activity, and degradation half-life are crucial for peptide therapeutics. The manuscript should at least discuss existing data on pantinin-1 stability and susceptibility to proteolytic degradation.

7. Result 3.1: As this result represents the inhibitory effect of pantinin-1 on purified CHIKV nsP2 protease. The main image showing the purified nsP2 protease is missing. Provide an image of the purified nsP2 protease (Coomassie gel image).

8. Figure 1A: It would be better to provide the confidence score of the alpha fold predicted image of Pantinin-1. Figure 1: 1B- Only the inhibitory effect of the compound is mentioned in the result section 3.1. whereas in Figure 1 B, the percentage of activity and inhibition both are depicted. Provide the detailed methods and results for the calculation of the activity percentage.

Figure 1c: Describe the methodology of IC50 in the Materials and Methods section.

9. Figure 2: The Author has mentioned N.control in both Figure 2A and 2B, but has not explained about this N.control, either in the result part or in the figure legend. In the figure legend, Triton X-100 was used as a positive control. Please specify the positive and negative controls and include them in the figure, the figure legend, and the respective results section.

10. The Author has mentioned that Pantinin-1 shows an inhibitory effect on the nsP2 protease by purifying the protease. To demonstrate its advantage in CHIKV viral replication, the author should assess antiviral activity in the presence of pantinin-1.

11. Methodology 2.7: The Author has selected the default parameter, but the parameter used in this study has not been explained properly.

12. Figures indicate triplicates, but it is unclear if these were biological or technical replicates. Provide replicates clearly and specify how many independent experiments were performed.

Minor Comments

1. The manuscript is mostly well written but would benefit from English editing. Several grammatical and typographical errors occur, e.g.: “Venom-derived peptides are new approach…” → “are a new approach”, “inhibition effect of the pantinin-1” → “inhibitory effect of pantinin-1”.

2. Figure legends should specify statistical tests, sample numbers (n), and p-values (wherever applicable).

3. Figure 1A (AlphaFold3 structure) should include confidence scores (pLDDT).

4. Specify whether nsP2pro was active-site verified (e.g., via autocleavage assay).

5. Clarify DMSO final concentration in enzyme assays; enzymatic activity can be sensitive.

6. Provide RMSD plots in the supplementary data for all poses.

7. Include docking scores or binding energy ranks before MD refinement to show initial selection criteria.

8. Ensure consistency in journal name abbreviations.

9. Table 3: Add a column indicating the experimental method used to assess inhibition (e.g., MST, FRET, cell-based).

The study is scientifically interesting and contributes to ongoing efforts to discover nsP2 protease inhibitors. However, significant experimental and interpretational strengthening is needed before the manuscript is ready for publication in PLOS ONE. The most critical improvement would be addition of viral cell culture assays.

6. PLOS authors have the option to publish the peer review history of their article (what does this mean?). If published, this will include your full peer review and any attached files.). If published, this will include your full peer review and any attached files.). If published, this will include your full peer review and any attached files.). If published, this will include your full peer review and any attached files.

...

Reviewer #1: No

Reviewer #2: No

---

## [Author Response · Author response to Decision Letter 1]

19 Jan 2026

Reviewer #1

This study investigates the inhibitory activity of pantinin-1, a venom-derived antimicrobial peptide from Pandinus imperator, against the nonstructural protein 2 protease (nsP2pro) of Chikungunya virus (CHIKV), a key enzyme in viral replication. Using a FRET-based assay, pantinin-1 was shown to inhibit nsP2pro in vitro with an IC₅₀ of 6.4 μM. Kinetic analysis confirmed a competitive inhibition mechanism, while biolayer interferometry determined a dissociation constant (KD) of 9.29 μM. Cytotoxicity assays in HEK 293 and Vero cells revealed no toxicity up to 20 μM, with moderate effects at 40 μM and higher. In silico molecular docking and molecular dynamics simulations supported binding at the protease active site, identifying key hydrophobic residues (L3, W7, F10, I13, V14) and substrate-binding residues (Y1079, Y1047, W1084) as critical for interaction.

The results demonstrate that pantinin-1 exerts potent inhibitory activity against CHIKV nsP2pro within a concentration range below the cytotoxic threshold. These findings suggest that pantinin-1 may serve as a lead compound for developing antiviral agents against CHIKV. As the authors noted, further investigations, particularly in virus-infected cell models, are required to validate its antiviral activity. The experiments are properly designed and executed, and the conclusions are convincingly supported by the data. As a reviewer, I have no critical comments on the manuscript and recommend its publication in PLOS One.

Response 1

Thank you for reviewing and evaluating our manuscript.

Reviewer #2

The manuscript titled “Inhibition of Chikungunya virus nsP2 protease in vitro by scorpion venom peptide pantinin-1” investigates the antiviral potential of pantinin-1, a 14-amino-acid antimicrobial peptide derived from Pandinus imperator. Using a series of biochemical and biophysical assays, the authors demonstrate that pantinin-1 inhibits CHIKV nsP2 protease activity with an IC₅₀ of 6.4 ± 2.04 µM and shows competitive inhibition behaviour. Biolayer interferometry reveals a KD of ~9 µM, suggesting direct binding to nsP2pro. Cytotoxicity assays show no major toxicity up to 20 µM. Molecular docking and MD simulations identify a putative binding mode at the substrate-binding pocket. Overall, the manuscript presents a novel venom-derived peptide as a promising nsP2pro inhibitor and provides a mix of experimental and computational evidence supporting its potential as an antiviral lead.

Major Comments

Comment 1

The concept of targeting nsP2 protease is well established, and the authors correctly highlight the scarcity of potent inhibitors. Demonstrating inhibition by pantinin-1 is novel and relevant. However, the manuscript would benefit from a clearer articulation of how pantinin-1 advances the field beyond prior peptide inhibitors (e.g., P1 peptide, RA-0002034). At present, the novelty seems incremental.

Response 1

We thank the reviewer for this important observation and agree that nsP2 protease is a well-validated antiviral target. While previous studies have reported peptide-based inhibitors such as P1 and small molecules including RA-0002034, pantinin-1 advances the field in several meaningful ways. First, pantinin-1 is a naturally occurring venom-derived peptide, representing a distinct and underexplored chemical space compared with synthetic peptides or repurposed compounds previously described. Importantly, it is a short 14–amino acid peptide, considerably smaller than earlier peptide inhibitors, which offers practical advantages for synthesis, structural optimization, and structure–activity relationship studies.

Beyond its origin and size, our work positions pantinin-1 as a new lead scaffold rather than a standalone inhibitor. Venom-derived peptides are biologically evolved molecules that have historically served as successful starting points for drug development, and their application to CHIKV nsP2 protease has not been previously reported. In addition, our study provides an integrated biochemical and biophysical characterization, including enzymatic inhibition, competitive binding kinetics (KD), cytotoxicity profiling, and in silico interaction analysis. This comprehensive approach strengthens the mechanistic understanding of nsP2pro inhibition and establishes a clear framework for rational optimization of pantinin-1 derivatives with improved potency and reduced toxicity.

Taken together, the novelty of this study lies not only in demonstrating nsP2 protease inhibition, but in introducing a venom-derived peptide scaffold with favorable size and tractability, thereby expanding the antiviral discovery landscape and providing a platform for future development of CHIKV therapeutics.

We included the following text in the manuscript introduction, “While nsP2 protease inhibitors have been previously described, pantinin-1 advances the field by introducing a venom-derived peptide from an unexplored chemical space. Its compact size, defined binding mode, and inhibitory activity within a non-cytotoxic concentration range position pantinin-1 as a promising starting point for further antiviral optimization.”

Comment 2

A major limitation is the absence of direct antiviral assays (e.g., CHIKV infection in Vero/BHK cells). Since cytotoxicity data are available, inclusion of EC₅₀ values (or at least plaque-reduction and viral RNA quantification) would substantially strengthen the conclusions. Without this, it is difficult to evaluate whether the protease inhibition observed in vitro translates into antiviral activity.

Response 2

We thank the reviewer for this important and well-taken comment. We fully agree that direct antiviral assays, such as CHIKV infection studies in Vero or BHK cells and determination of EC₅₀ values, would substantially strengthen the translational relevance of our findings. The primary objective of the present study, however, was to establish proof-of-concept biochemical validation of pantinin-1 as an inhibitor of CHIKV nsP2 protease and to characterize its mechanism of action through enzymatic, kinetic, binding, and in silico analyses.

While cytotoxicity data demonstrate that pantinin-1 inhibits nsP2pro at concentrations below that causing significant cell toxicity, we acknowledge that enzymatic inhibition does not necessarily translate directly into antiviral efficacy in cell-based systems. Further optimization of the pantinin-1 sequence and evaluation of antiviral activity, including plaque-reduction assays and viral RNA quantification, is therefore a critical next step and is currently planned as part of our ongoing work. We have revised the manuscript to explicitly acknowledge this limitation and to frame pantinin-1 as a lead scaffold rather than a validated antiviral agent, emphasizing the need for future cell-based and in vivo studies to confirm its antiviral potential.

We added therefore the following text to the conclusion and outlook section, “The present work establishes pantinin-1 as a biochemical inhibitor of CHIKV nsP2 protease. After further optimization of the peptide, the determination of EC₅₀ values using plaque-reduction and viral RNA assays will be essential to confirm its potential as an antiviral lead.”

Comment 3

The Lineweaver–Burk analysis suggests competitive inhibition, but these double-reciprocal plots are prone to error and are no longer recommended as standalone mechanistic proof. Consider supplementing with nonlinear global fitting of Michaelis–Menten data/enzyme progress curve analysis.

Response 3

Thank you for the suggestion, the lineweaver-burk analysis is to see in combination with our docking and MD approach. In both cases the results show, that Pantinin-1 act as a competitive inhibitor. Especially the docking and MD approach demonstrated, that the peptide bind near or within the substrate-binding site, which could block this region for the interaction with the substrate. An inhibitor that bind in this substrate-binding site stand in competition with the substrate and is therefore a competitive inhibitor. However, we added a new sub-figure (1C), in figure 1, showing the nonlinear global fitting of Michaelis-Menten enzyme/inhibitor progress curve. Which even shows the typical characteristics of a competitive inhibitor by increasing the Km value. Which is described for several inhibitors with a competitive inhibition mode, please have a look in the following publications:

• Lee et al. 2024. https://doi.org/10.3390/biom14010099

• Vazquez-Armenta et al. 2025. https://doi.org/10.3390/catal15030257

• Coria et al. 2016. https://doi.org/10.4049/jimmunol.1501188

• Venditti et al. 2013. https://doi.org/10.1021/cb400027q

• Sorribas, Guillén and Sosa 2019. https://doi.org/10.1007/s00424-018-2241-x

We revised the results section and the discussion section accordingly.

Comment 4

The MD simulations reveal stable poses, but the conclusion that pantinin-1 is a competitive inhibitor because it binds the active site is not fully convincing. The final MD pose appears shifted from the catalytic dyad, raising questions about how substrate exclusion occurs mechanistically. The authors should explicitly compare the peptide binding pose with the native substrate (P1234) cleavage motif or model the peptide and substrate simultaneously. Per-residue decomposition identifies strong hydrophobic contributions, but experimental validation (e.g., alanine-scanning) is essential before stating these as "key binding residues."

Response 4

We are grateful for this comment, as mentioned in response 3, the docking and MD approach demonstrated, that pantinin-1 bind near or within the substrate-binding site, which block this region for the interaction with the substrate. However, the predicted binding mode of pantinin-1, although leaving the catalytic residues exposed, hinders the binding of a substrate peptide at the Pn side. This result is presented in the SI of the current version (S6 Fig). In addition, we have added the following sentences to the manuscript results section:

“However, the predicted binding mode is still capable of sterically hindering substrate access to the active site. This is supported by the superposition of nsP2pro in complex with pantinin-1 and the crystal structure of the Venezuelan equine encephalitis virus (VEEV) nsP2pro containing an N-terminal segment in a substrate-like conformation within the active site (S6 Fig).”

Regarding the per-residue decomposition results, we have clearly stated in the manuscript that the predicted free energy contribution values are a prediction based on the proposed binding mode of pantinin-1. It is beyond the scope of the manuscript to perform alanine scanning to validate the predictions.

Comment 5

Although the peptide is non-toxic up to 20 µM, its IC50 is 6.4 µM, leaving a narrow therapeutic window. The discussion should compare this with the typical requirements for peptide antiviral drugs and address potential strategies to reduce cytotoxicity (e.g., peptide engineering, truncation, amino acid modification, etc.).

Response 5

We thank the reviewer for highlighting this important point. We acknowledge that the current IC₅₀ value of pantinin-1 (6.4 µM) relative to its non-toxic concentration range suggests a modest therapeutic window, which is not uncommon for early-stage peptide leads. Importantly, many peptide antivirals undergo substantial optimization to improve selectivity and reduce cytotoxicity, and pantinin-1’s short length and defined structure make it well suited for such engineering strategies, including residue substitution, charge modulation, truncation, and incorporation of non-natural amino acids. We have revised the Discussion to place our findings in the context of peptide antiviral development and to explicitly outline these optimization approaches.

We added the following text to the discussion section,

“Peptide-based antiviral leads often exhibit low micromolar inhibitory activity at early stages of development, frequently accompanied by a relatively narrow therapeutic window that necessitates subsequent optimization [71-73]. The amphipathic and α-helical nature of pantinin-1, while contributing to its bioactivity, may also underlie the observed cytotoxicity at higher concentrations, a characteristic commonly reported for membrane-active and venom-derived peptides [74–76]. Importantly, such limitations can be mitigated through peptide engineering strategies, including sequence truncation, charge redistribution, reduction of hydrophobicity, and incorporation of non-natural or D-amino acids to improve selectivity and safety profiles [77-79]. Given its short length and defined structure, pantinin-1 represents a suitable scaffold for rational optimization aimed at enhancing its therapeutic index. Similar optimization approaches have successfully advanced other peptide antivirals from micromolar leads to clinically relevant candidates [80,81].”

Comment 6

Protease resistance, serum stability, haemolytic activity, and degradation half-life are crucial for peptide therapeutics. The manuscript should at least discuss existing data on pantinin-1 stability and susceptibility to proteolytic degradation.

Response 6

We added the following information to the manuscript discussion section.

“Pantinin-1 and -2, isolated from the scorpion Pandinus imperator, are known to have activity against Gram-positive bacteria, fungi, and viruses. Beside this broad antimicrobial impact, it was shown that Pantinin-1 displays substantial stability in human serum and a strong resistance to proteolytic degradation. After 16 hours of incubation, approximately 70–80% of the peptide remained intact. Furthermore, Pantinin-1 caused no significant hemolysis of human erythrocytes at concentrations up to 512 µg/mL. While these results indicate the potential for an extended biological half-life, detailed in vivo pharmacokinetic data have yet to be reported [56-59].”

• Giugliano et al. 2025. https://doi.org/10.3389/fmicb.2025.1569719

• Giugliano et al. 2025. https://doi.org/10.3390/pathogens14070713

• Giugliano et al. 2025. https://doi.org/10.1002/cmdc.202500333

• Capasso et al. 2025. https://doi.org/10.3390/microorganisms14010068

Comment 7

Result 3.1: As this result represents the inhibitory effect of pantinin-1 on purified CHIKV nsP2 protease. The main image showing the purified nsP2 protease is missing. Provide an image of the purified nsP2 protease (Coomassie gel image).

Response 7

We thank the reviewer for bringing up this important point. We have now added an image of a Coomassie stained SDS PAGE of purified nsP2 protease. The image has been included in the supplementary material (S1 Fig.).

Comment 8

Figure 1A: It would be better to provide the confidence score of the alpha fold predicted image of Pantinin-1. Figure 1: 1B- Only the inhibitory effect of the compound is mentioned in the result section 3.1. whereas in Figure 1 B, the percentage of activity and inhibition both are depicted. Provide the detailed methods and results for the calculation of the activity percentage.

Figure 1c: Describe the methodology of IC50 in the Materials and Methods section.

Response 8

We included the AlphaFold confidence value of the predicted structure (>90%) in Fig. 1.

Comment 9

Figure 2: The Author has mentioned N.control in both Figure 2A and 2B, but has not explained about this N.control, either in the result part or in the figure legend. In the figure legend, Triton X-100 was used as a positive control. Please specify the positive and negative controls and include them in the figure, the figure legend, and the respective results section.

Response 9

Thank you for the comment; we changed N. control to Triton X-100 in Figure 2 accordingly and uploaded the revised figure.

Comment 10

The Author has mentioned that Pantinin-1 shows an inhibitory effect on the nsP2 protease by purifying the protease. To demonstrate its advantage in CHIKV viral replication, the author should assess antiviral activity in the presence of pantinin-1.

Response 10

Thank you for the comment, but as already mentioned in the response to the main comment 2, we agree with the reviewer, that direct antiviral assays, including CHIKV infection experiments in Vero or BHK cells and determination of EC₅₀ values, would cons

---

## [Decision Letter · Decision Letter 1]

13 Feb 2026

PONE-D-25-43080R1Inhibition of Chikungunya virus nsP2 protease in vitro by scorpion venom peptide pantinin-1PLOS One

Dear Dr. Eberle,

Thank you for submitting your manuscript to PLOS ONE. After careful consideration, we feel that it has merit but does not fully meet PLOS ONE’s publication criteria as it currently stands. Therefore, we invite you to submit a revised version of the manuscript that addresses the points raised during the review process.

Based on the reviewer's comments, the infection study will add greater value to the manuscript. In vitro studies involving CHIKV infection in cell lines are relevant to translatability. If you do not have this data, please let us know and provide a rebuttal for assessment.

We look forward to receiving your revised manuscript.

Kind regards,

Daniel Parkes, PhD

Staff Editor

PLOS One

On behalf of

Seth Agyei Domfeh, PhDAcademic EditorPLOS One

Journal Requirements:

Reviewers' comments:

Reviewer's Responses to Questions

**Comments to the Author**

1. If the authors have adequately addressed your comments raised in a previous round of review and you feel that this manuscript is now acceptable for publication, you may indicate that here to bypass the “Comments to the Author” section, enter your conflict of interest statement in the “Confidential to Editor” section, and submit your "Accept" recommendation.

Reviewer #1: All comments have been addressed

Reviewer #2: (No Response)

2. Is the manuscript technically sound, and do the data support the conclusions?

Reviewer #1: Yes

Reviewer #2: Partly

3. Has the statistical analysis been performed appropriately and rigorously? 

Reviewer #1: Yes

Reviewer #2: Yes

4. Have the authors made all data underlying the findings in their manuscript fully available?

The PLOS Data policy requires authors to make all data underlying the findings described in their manuscript fully available without restriction, with rare exception (please refer to the Data Availability Statement in the manuscript PDF file). The data should be provided as part of the manuscript or its supporting information, or deposited to a public repository. For example, in addition to summary statistics, the data points behind means, medians and variance measures should be available. If there are restrictions on publicly sharing data—e.g. participant privacy or use of data from a third party—those must be specified.requires authors to make all data underlying the findings described in their manuscript fully available without restriction, with rare exception (please refer to the Data Availability Statement in the manuscript PDF file). The data should be provided as part of the manuscript or its supporting information, or deposited to a public repository. For example, in addition to summary statistics, the data points behind means, medians and variance measures should be available. If there are restrictions on publicly sharing data—e.g. participant privacy or use of data from a third party—those must be specified.requires authors to make all data underlying the findings described in their manuscript fully available without restriction, with rare exception (please refer to the Data Availability Statement in the manuscript PDF file). The data should be provided as part of the manuscript or its supporting information, or deposited to a public repository. For example, in addition to summary statistics, the data points behind means, medians and variance measures should be available. If there are restrictions on publicly sharing data—e.g. participant privacy or use of data from a third party—those must be specified.requires authors to make all data underlying the findings described in their manuscript fully available without restriction, with rare exception (please refer to the Data Availability Statement in the manuscript PDF file). The data should be provided as part of the manuscript or its supporting information, or deposited to a public repository. For example, in addition to summary statistics, the data points behind means, medians and variance measures should be available. If there are restrictions on publicly sharing data—e.g. participant privacy or use of data from a third party—those must be specified.

Reviewer #1: Yes

Reviewer #2: Yes

5. Is the manuscript presented in an intelligible fashion and written in standard English?

Reviewer #1: Yes

Reviewer #2: Yes

6. Review Comments to the Author

Reviewer #1: I have no critical comments on the manuscript. The study is well designed, clearly presented, and the results are sound and adequately discussed; therefore, I recommend its publication in PLOS One.

Reviewer #2: The responses submitted by the authors are satisfactory. However, CHIKV infection studies in at least Vero cells in the presence of pantinin-1 (including appropriate controls) and determination of EC₅₀ values would substantially boost the translational relevance of the findings. It would definitely add more value to the manuscript.

7. PLOS authors have the option to publish the peer review history of their article (what does this mean?). If published, this will include your full peer review and any attached files.). If published, this will include your full peer review and any attached files.). If published, this will include your full peer review and any attached files.). If published, this will include your full peer review and any attached files.

...

Reviewer #1: No

Reviewer #2: **Yes:**Dr. Soma ChattopadhyayDr. Soma ChattopadhyayDr. Soma ChattopadhyayDr. Soma Chattopadhyay

---

## [Author Response · Author response to Decision Letter 2]

27 Feb 2026

Response to the Reviewer

Reviewer #1

I have no critical comments on the manuscript. The study is well designed, clearly presented, and the results are sound and adequately discussed; therefore, I recommend its publication in PLOS One.

Response Reviewer#1

Thank you for reviewing and evaluating our manuscript.

Reviewer #2

The responses submitted by the authors are satisfactory. However, CHIKV infection studies in at least Vero cells in the presence of pantinin-1 (including appropriate controls) and determination of EC₅₀ values would substantially boost the translational relevance of the findings. It would definitely add more value to the manuscript.

Response Reviewer#2

We sincerely thank the reviewer for this thoughtful and constructive comment. We fully agree that in vitro infection studies involving CHIKV-infected cell lines would substantially enhance the translational value of the manuscript.

At this stage, however, we respectfully decline to include CHIKV infection experiments. The primary objective of the present study was to identify and characterize a lead peptide candidate and determine its initial in vitro activity (IC₅₀) under controlled conditions. We consider this an essential first step in a structured drug development process.

Importantly, the current peptide still requires further optimization to improve potency, metabolic stability, cell uptake and overall pharmacological properties. We believe that performing infection-based experiments prior to sequence optimization would be premature and may not fully reflect the therapeutic potential of an improved candidate. Our immediate focus is therefore on rational peptide refinement to enhance efficacy. Once an optimized version is obtained, we fully agree that evaluation in CHIKV infection models—followed by in vivo studies—will be scientifically justified and necessary.

We would also like to emphasize the broader context of this work. Chikungunya virus is expanding geographically, and Europe itself is increasingly at risk of outbreaks. We do not claim to have developed a definitive therapeutic solution. Rather, we present a promising peptide scaffold that demonstrates measurable in vitro activity and represents a credible starting point for further development. With continued optimization, this candidate could progress toward more advanced preclinical evaluation.

To address the reviewer’s concern, we have clarified in the revised manuscript (Conclusion and Outlook section) that:

1. The current findings represent an early-stage lead identification effort.

2. Peptide optimization is ongoing.

3. Infection-based and in vivo studies are planned as future steps once an improved candidate is available.

We respectfully hope the reviewer and the editorial office will consider that the present manuscript provides a solid and scientifically sound foundation for subsequent translational studies. We sincerely appreciate the reviewer’s valuable input, which has helped us better position the scope and future direction of our work.

---

## [Decision Letter · Decision Letter 2]

26 Mar 2026

Inhibition of Chikungunya virus nsP2 protease in vitro by scorpion venom peptide pantinin-1

PONE-D-25-43080R2

Dear Dr. Eberle,

We’re pleased to inform you that your manuscript has been judged scientifically suitable for publication and will be formally accepted for publication once it meets all outstanding technical requirements.

Kind regards,

Seth Agyei Domfeh, PhD

Academic Editor

PLOS One

Additional Editor Comments (optional):

Reviewers' comments:

Reviewer's Responses to Questions

**Comments to the Author**

1. If the authors have adequately addressed your comments raised in a previous round of review and you feel that this manuscript is now acceptable for publication, you may indicate that here to bypass the “Comments to the Author” section, enter your conflict of interest statement in the “Confidential to Editor” section, and submit your "Accept" recommendation.

Reviewer #2: All comments have been addressed

2. Is the manuscript technically sound, and do the data support the conclusions?

Reviewer #2: Yes

3. Has the statistical analysis been performed appropriately and rigorously? 

Reviewer #2: Yes

4. Have the authors made all data underlying the findings in their manuscript fully available?

The PLOS Data policy requires authors to make all data underlying the findings described in their manuscript fully available without restriction, with rare exception (please refer to the Data Availability Statement in the manuscript PDF file). The data should be provided as part of the manuscript or its supporting information, or deposited to a public repository. For example, in addition to summary statistics, the data points behind means, medians and variance measures should be available. If there are restrictions on publicly sharing data—e.g. participant privacy or use of data from a third party—those must be specified.requires authors to make all data underlying the findings described in their manuscript fully available without restriction, with rare exception (please refer to the Data Availability Statement in the manuscript PDF file). The data should be provided as part of the manuscript or its supporting information, or deposited to a public repository. For example, in addition to summary statistics, the data points behind means, medians and variance measures should be available. If there are restrictions on publicly sharing data—e.g. participant privacy or use of data from a third party—those must be specified.requires authors to make all data underlying the findings described in their manuscript fully available without restriction, with rare exception (please refer to the Data Availability Statement in the manuscript PDF file). The data should be provided as part of the manuscript or its supporting information, or deposited to a public repository. For example, in addition to summary statistics, the data points behind means, medians and variance measures should be available. If there are restrictions on publicly sharing data—e.g. participant privacy or use of data from a third party—those must be specified.requires authors to make all data underlying the findings described in their manuscript fully available without restriction, with rare exception (please refer to the Data Availability Statement in the manuscript PDF file). The data should be provided as part of the manuscript or its supporting information, or deposited to a public repository. For example, in addition to summary statistics, the data points behind means, medians and variance measures should be available. If there are restrictions on publicly sharing data—e.g. participant privacy or use of data from a third party—those must be specified.

Reviewer #2: Yes

5. Is the manuscript presented in an intelligible fashion and written in standard English?

Reviewer #2: Yes

6. Review Comments to the Author

Reviewer #2: Satisfactory explanations have been provided for all the queries. Authors did not perform all the suggested experiments, however provided explanations for not conducting those experiments.

7. PLOS authors have the option to publish the peer review history of their article (what does this mean?). If published, this will include your full peer review and any attached files.). If published, this will include your full peer review and any attached files.). If published, this will include your full peer review and any attached files.). If published, this will include your full peer review and any attached files.

...

Reviewer #2: No

---

## [Editor Report · Acceptance letter]

PONE-D-25-43080R2

PLOS One

Dear Dr. Eberle,

I'm pleased to inform you that your manuscript has been deemed suitable for publication in PLOS One. Congratulations! Your manuscript is now being handed over to our production team.

Kind regards,

on behalf of

Dr. Seth Agyei Domfeh

Academic Editor

PLOS One